# Influence of Fasting until Noon (Extended Postabsorptive State) on Clock Gene mRNA Expression and Regulation of Body Weight and Glucose Metabolism

**DOI:** 10.3390/ijms24087154

**Published:** 2023-04-12

**Authors:** Daniela Jakubowicz, Rachel Chava Rosenblum, Julio Wainstein, Orit Twito

**Affiliations:** Endocrinology and Diabetes Unit, Wolfson Medical Center, Sackler Faculty of Medicine, Tel Aviv University, Holon 58100, Israel

**Keywords:** fasting until noon, circadian clock genes, fasting until noon, weight loss, overall glycemia, diet induced thermogenesis

## Abstract

The trend of fasting until noon (omission or delayed breakfast) is increasingly prevalent in modern society. This eating pattern triggers discordance between endogenous circadian clock rhythms and the feeding/fasting cycle and is associated with an increased incidence of obesity and T2D. Although the underlying mechanism of this association is not well understood, growing evidence suggests that fasting until noon, also known as an “extended postabsorptive state”, has the potential to cause a deleterious effect on clock gene expression and to disrupt regulation of body weight, postprandial and overall glycemia, skeletal muscle protein synthesis, and appetite, and may also lead to lower energy expenditure. This manuscript overviews the clock gene-controlled glucose metabolism during the active and resting phases and the consequences of postponing until noon the transition from postabsorptive to fed state on glucose metabolism, weight control, and energy expenditure. Finally, we will discuss the metabolic advantages of shifting more energy, carbohydrates (CH), and proteins to the early hours of the day.

## 1. Introduction

The endogenous circadian clock temporally coordinates the diurnal variation of most metabolic processes to anticipate the daily changes in the light/dark cycle and nutrient availability to adjust the eating and fasting schedule to the optimal time of the day [1,2,3,4,5,6].

The circadian coordination requires an alignment between the central clock in the hypothalamic suprachiasmatic nucleus (SCN), which is responsive to light inputs, with the peripheral clock genes disseminated in almost all the tissues (i.e., β-cells, muscle, liver), mostly entrained to the hour of food intake [1,2,3,4,5,6,7,8,9,10].

Evidence suggests that the metabolic conditions for food intake are optimal in the morning [11,12,13,14,15,16,17,18]. Specifically, the first meal of the day, consumed at the beginning of the activity phase (early-timed breakfast), is a powerful “zeitgeber”, or starter of peripheral clock genes expression [4,8,9,19,20,21,22]. It enhances enzymes and hormones involved in the regulation of body weight [19,23,24,25], overall glycemia [18,26,27,28,29], muscle synthesis [30,31,32], and appetite [12,25,33]. Further, the diet-induced thermogenesis is significantly higher after a high-energy breakfast vs. an isocaloric dinner, highlighting the importance of the first meal of the day in achieving metabolic homeostasis [25,34,35,36,37].

While many studies documented the metabolic advantage of early-timed breakfast [17,18,19,20,21,22,23,24,25,26], growing evidence shows that the “lack of breakfast” or delayed breakfast until noon, also known as the “extended postabsorptive state” the transition from overnight fast to fed state, has the potential to desynchronize clock gene expression and regulation of metabolism [19,20,21,22,38]. The omission of breakfast has been associated with higher body mass indices, insulin resistance, hyperglycemia, and the risk for developing T2D [19,23,28,39,40,41,42,43]. The circadian misalignment and disrupted clock gene expression arising from fasting until noon leads to increased glycemia and appetite scores, deficient insulin and incretin responses after subsequent meals [19,20,21,22,23,24,25,27,38,44,45,46], and diminished energy expenditure [25,34,35,36,37]. Further, the lack of breakfast activates muscle protein breakdown to provide the liver with glucogenic amino acids for neoglucogenesis, which may result in a loss of skeletal muscle mass [30,31,32,41,45,46,47].

This review aims to discuss the circadian misalignment derived from the lack of breakfast and the ensuing deleterious effects on clock genes expression, body weight, glucose metabolism, muscle mass, and energy expenditure. We will provide a general overview of the circadian clock molecular mechanism and its role in aligning metabolic processes to the active and resting phases. The clock-controlled postprandial and postabsorptive glucose metabolism and the transition or switch from overnight fast to a fed state. Finally, we will discuss what time of the day is most appropriate for fasting and eating to achieve optimal control of body weight, energy expenditure, and glucose metabolism.

## 2. Circadian Clock Molecular Mechanism

The molecular mechanism of central and peripheral clocks consists of transcriptional/translational positive and negative feedback loops coordinating an extensive clock gene network to maintain a self-sustained daily ~24 h oscillation [1,2,3]. The clock gene set includes *BMAL1* (brain and muscle ARNT-like 1), *CLOCK* (circadian locomotor output cycles kaput), three period genes (*PER1*, PER2, and PER3), two cryptochrome genes (*CRY1* and *CRY2*), the reverse erythroblastosis virus (*REV-ERB* α, β, δ, and γ), and retinoic acid-related orphan receptor (*ROR* α, β, and γ) [2,3,4,5,6,7,48,49,50,51,52,53,54,55].

The transcriptional activators *CLOCK* and *BMAL1* dimerize, forming the CLOCK: BMAL1 complex, which associates with *Sirtuin 1* (*SIRT1*) deacetylase. CLOCK: BMAL1 complex rhythmically binds E-box-like promoter sequences to activate the transcription of *PER* and *CRY* repressor genes. The resulting PER and CRY proteins interact to form PER: CRY dimers in the cytoplasm. Upon reaching a critical threshold concentration in the cytoplasm, PER: CRY dimers translocate back to the nucleus, repressing their own CLOCK: BMAL1 transcription [1,2,3,7,48,49,50,51,52,53,54,55]. The blockage of CLOCK: BMAL1 transcription persists until all PER: CRY dimers are degraded, and then the cycle begins again, thus maintaining the ~24 h oscillation. The blockage of CLOCK: BMAL1 is reversed by the action of casein kinase I epsilon (CKIε), allowing the resumption of a new ~24 h transcription cycle. Further, CKIε also phosphorylates and partially reactivates the *BMAL1-driven* transcription [1,48,49].

In another major transcriptional loop, CLOCK: BMAL1 complex encodes the transcription of repressor proteins *REV-ERBs* and the activators *RORs*, (α, β, and γ), forming an additional stabilizing loop with *BMAL1*, further maintaining a ~24 h rhythm in *BMAL1* transcription [1,2,3,4,5,6,7,50,51,52,53,54,55,56].

The oxidized form NAD+ activates *SIRT1 and* interacts with *CLOCK*: *BMAL1* in a circadian manner, promoting the *deacetylation* and degradation of *PER2* and is a crucial factor in glucose metabolism [3,7,49,50,51,52,53,54,55]. In addition, the adenosine monophosphate-activated protein kinase (AMPK) positively interacts with *SIRT1* [56] Figure 1.

Another subset of clock-controlled genes (CCGs), transcriptional factors, and clock proteins include all members of the PPAR family [α, β, γ, δ] and PPARγ coactivator 1α (PGC-1α), which feed back into the clock, producing additional bidirectional regulation of metabolism and circadian rhythms [1,53,54,55]. Hence, *CLOCK: BMAL1*-driven transcription of *PERs*, *CRYs*, *REV-ERBα*, *RORα* genes, and PGC-1α promote the expression of tissue-specific clock-controlled genes, activating many rhythmic metabolic and hormonal outputs [56,57,58,59,60,61,62,63,64,65,66,67,68]. The expression of *BMAL-1*, *RORα* and *SIRT1* during the active phase (fed state) is essential in the regulation of the circadian changes of insulin sensitivity [7,60,61], muscular GLUT-4 activity, muscle glucose uptake [60,68], and the β-cells responsiveness [58,63,64,65]. The circadian incretin, i.e., glucagon-like peptide-1 (GLP-1) postprandial secretion in the intestinal L-cells, also relies on *BMAL-1* and *RORα* integrity [65,66]. Adenosine monophosphate-activated protein kinase (*AMPK*) exerts a positive effect on *SIRT1* and further enhances β-cells viability, GLUT-4 expression, and muscular glucose uptake [7,60,61].

During the nocturnal resting phase, the expression of *REVERBα*, *RORα*, and *SIRT1* in the liver promotes glycogenolysis enzyme glucose 6-phosphatase (HG6-P) in the first part of the nocturnal resting phase and phosphoenolpyruvate carboxykinase (PEPCK) in the gluconeogenesis pathway, namely during the second part of the nocturnal phase [56,57]. In addition, *BMAL1* transcription of *PPARα* and *PGC-1α* in the adipose tissue plays a role in lipid metabolism (lipogeneses and nocturnal lipolysis [52,55] Figure 1).

## 3. Synchrony between Central and Peripheral Clock Genes

The light-entrained central clock in SCN coordinates peripheral clocks through sympathetic-pathway signals, setting the sleep–wake cycle and hormonal signaling, primarily adrenal glucocorticoids [1,2,3,4,7]. This alignment occurs at a “specific time of the day”, leading to an appropriate temporal sequence of hormonal, digestive, absorptive, and metabolic functions [5,7,8,9,10,49,50].

Thus, the clock timing system adjusts to the active and resting phase, the oscillation of hepatic glucose output [56,58], ghrelin secretion [59], insulin sensitivity, and β-cell responsiveness [58,60,61,62,63,64]. It also regulates the diurnal variation of the postprandial secretion of incretins, i.e., GLP-1 and glucagon-like peptide-2 (GLP-2) in the intestinal L-cells, gastric inhibitory polypeptide (GIP) in intestinal K-cells [65,66], and leptin and adiponectin secretion in the adipose tissue [55,67]. Notably, the clock genes in skeletal muscle regulate GLUT-4 activity, muscle glucose uptake [60,68], muscle protein synthesis during the postprandial state, and muscle protein breakdown in the postabsorptive state [30,31,32,41]. Figure 2.

## 4. Circadian Variation of Metabolism during Active and Resting Phase

The coordination between central and peripheral clocks generates the diurnal variation of the metabolic processes throughout the active and resting phases to ensure the consumption of “the right meals at the right and optimal time of the day” [5,49,69,70].

### 4.1. Clock Controlled Active and Resting Phase

The diurnal variation of melatonin and cortisol, and the sleep–wake cycle driven by the light/dark controlled SCN clock, split the daily metabolic outputs into the active phase (biological day) and resting phase (biological night) [71]. The active phase in humans begins upon waking up between 06:00 and 08:00 and continues until 22:00 when the nocturnal rise of melatonin initiates the resting phase [6,72,73]. The morning peak of cortisol is an important starting signal. It prepares the body for waking up and consuming the first meal of the day (i.e., breakfast) to cover the increase in the energetic demands of the active phase [74].

### 4.2. Clock Controlled Metabolism Is Enhanced in the Early Hours of the Active Phase

The clock-gene regulation enhances most metabolic pathways at dawn, in the early hours of the active phase, and shortly after waking up [8,9,10,11,12,13,14,15,16,17,18,69,70]. The secretion of ghrelin peaks at 08:00, increasing appetite and food intake in the early morning [59]. Adiponectin levels also rise in the morning, peaking at 11:00 and declining at 20:00 [74,75,76]. The morning rise of adiponectin stimulates via activation of AMPK, the fatty acid oxidation, insulin sensitivity, muscle glucose uptake, and glycolysis while reducing hepatic glucose production during the morning hours. These mechanisms increase glucose utilization and decrease fat accumulation in the early hours suggesting fewer fattening effects from the meal ingested in the early morning [6,74,75,76,77]. The decreased adiponectin in the evening induces metabolic change toward insulin-mediated anabolic processes. Thus, the postprandial insulin response in the evening stimulates fat storage and lipogenesis by activating fatty acid synthesis [75,76,77]. In contrast, the rise of leptin in the evening reduces fat accumulation and increases nocturnal lipolysis [74].

The clock gene regulation also enhances insulin sensitivity, β-cell responsiveness, GLUT-4 activity, muscle glucose uptake, post-meal incretin (i.e., GLP-1, GLP-2, and *GIP)* secretion, and incretin-mediated insulinotropic effect in the early hours of the active phase [58,60,61,62,63,64,65,66,68]. As a result, the glycemic response is significantly higher after identical meals consumed in the evening compared to the morning [18,36,58,78,79,80].

The resting energy expenditure (REE) also displays circadian oscillation showing its lowest levels during the resting phase, while the respiratory quotient (RQ), reflecting macronutrient utilization, is at its highest score in the early hours of the active phase [81]. Moreover, the circadian clock enhances diet-induced thermogenesis (DIT) or energy expenditure after meals consumed in the early active phase compared to the evening [34,35,36,37]. Therefore, the early hours of the active phase (i.e., breakfast) are optimal for food intake and especially for CH consumption, while in the evening and nighttime, it is likely more convenient to reduce the energy and CH intake [8,9,10,11,12,13,14,15,16,17,18,69,70,77,78,79,80,81].

### 4.3. Potential Benefits of Early-Timed Breakfast

Feeding/fasting misalignment, i.e., small breakfast or breakfast omission and higher caloric intake in the evening, e.g., in shift workers, is associated in preclinical and clinical studies with circadian clock disruption [19,22] and increased risk of developing obesity, hyperglycemia, and diabetes [16,17,18,19,20,21,22,38,39,40]. Studies on time-restricted eating (TRE) consisting of an eating window restricted to the early hours of the day (eTRE) and fasting during the late hours and night are associated with improved regulation of weight loss and overall glucose levels compared to late TRE with extended fasting until noon and an eating window in the afternoon and evening [8,9,10,11,12,15,16,77,79,80]

In participants at risk for T2D, a late TRE consisting of skipping breakfast and eating from 12:00 to 21:00 was compared to an early TRE with early-timed breakfast and an eating window from 8:00 to 17:00. The early TRE led to lower mean fasting glucose by continuous glucose monitor versus baseline. The fasting triglycerides decreased in both groups, but there was no difference between early or late TRE conditions [79]. Another study in healthy men compared an early mealtime diet with three meals at 8:00, 13:00, and 18:00 to a late mealtime diet, skipping breakfast and eating at 13:00, 18:00, and 23:00, for two weeks [80]. Plasma triglycerides and total and LDL cholesterol levels were significantly decreased in the early mealtime diet suggesting that skipping breakfast has a deleterious effect on lipid metabolism [80].

Our group further explored whether shifting most energy and CH to the early hours of the day is an effective meal-timing approach for weight loss and reduction of overall daily glycemia. We compared the effects of a high-energy breakfast diet (Bdiet) versus a high-energy dinner diet (Ddiet) in overweight and obese women over 12 weeks [17]. The high-energy breakfast diet (Bdiet) consisted of (700 kcal breakfast, 500 kcal lunch, and 200 kcal dinner), while the high-energy dinner diet (Ddiet) had the reverse schedule (200 kcal breakfast, 500 kcal lunch, 700 kcal dinner). Bdiet led to more efficient weight loss, reduced overall daily glucose and insulin resistance, and lower hunger scores and ghrelin levels than Ddiet. Interestingly, mean triglyceride levels decreased by 33.6% in the Bdiet group but increased by 14.6% in the Ddiet [17]. In summary, greater weight loss and reduced glycemia and lipids were observed in the Bdiet group, with energy intake shifted towards early rather than late hours in the day [17].

## 5. Circadian Clock Regulation of Postprandial and Postabsorptive Glucose Metabolism and the Transition (Switch) from Overnight Fast to Postprandial Fed State

The alignment of the fasting/feeding cycle with the circadian clock engages dynamic feedback to the clock gene regulatory network. It regulates the insulin and glucagon-dependent metabolic tissues and the circadian oscillation of glucose metabolism throughout the postabsorptive and postprandial states and during the shift or transition from postabsorptive to postprandial or fed state at dawn [3,4,38,45].

In the active phase (postprandial state) which starts upon the consumption of the early-timed breakfast, the glucose and insulin response to the influx of dietary CH, initiates the translocation of GLUT4, the skeletal muscle glucose uptake, and the reduction of liver glucose production [38,82,83,84,85]. While, in the resting phase (postabsorptive state), the glucose metabolism consists of hepatic glucose production (glycogenolysis and gluconeogenesis) [3,4,38,45].

The expression of *BMAL-1*, *RORα*, *SIRT1*, and AMPK is essential during the fed or postprandial state for the regulation of the diurnal variation of β-cells responsiveness [58,62,63,64,65,66], insulin sensitivity [60,61], GLUT-4 activity, and muscle glucose uptake [60,68]. During the overnight fast (postabsorptive state), the expression of *REVERBα*, *RORα*, SIRT1, and *PGC-1α* in the liver is critical for the regulation of the nocturnal hepatic glucose production, namely glycogenolysis and gluconeogenic pathways, to maintain adequate blood glucose during overnight sleep [56,57,86]. Figure 3.

### 5.1. Clock Controlled Metabolism during Postprandial State

At dawn, shortly after waking up, the postprandial rise of glucose, insulin, and incretins (i.e., GLP-1) after breakfast upregulate the transcription of clock genes expression [21,22,38,81,82,83,84,85,86,87], improving insulin sensitivity, β-cells responsiveness [58,60,61,62,63,64], GLUT-4 activity, and muscle glucose uptake [60,68].

Thus, the early-timed breakfast enhances glycolysis and glycogen synthesis to use glucose as fuel and to replenish glycogen stores [38,82,83,84,85]. It also decreases adipose tissue lipolysis, reducing the blood-free fatty acids (FFA) and further facilitating insulin-stimulated glucose uptake during the remaining hours of the active phase [82,83].

At the molecular level, it has been documented that early-timed breakfast upregulates *CLOCK*, *BMAL1*, *RORα*, *SIRT1*, AMPK, and *PER2* clock gene expression [21,22,38,82,83,84,85,86,87] and several transcriptional factors, such as CH response element binding protein (*ChREBP*), that promote the activity of hexokinase (HK) and 6-phosphofructokinase (PFK), key enzymes in skeletal muscle glycolysis, enhancing glucose utilization as the primary fuel during postprandial state [38,82,83].

The early-timed breakfast positively influences the SRIT1-AMPK interaction, which further benefits insulin sensitivity, GLUT-4 translocation, and muscle glucose uptake [7,56,60,61,86,87,88,89], improving the glucose and insulin postprandial responses in the morning versus evening [7,56,86].

At the same time the increased *CLOCK* and PER2 expression activates the transcription of glycogen synthase 2 (GYS2) and hepatic glycogen synthesis [38,84,85]. Therefore, the excess of postprandial glucose is transported to the liver and stored as glycogen, while glycogenolysis is inhibited [38,84,85]. *BMAL-1* and *RORα* also regulate the diurnal variation of postprandial incretin (i.e., GLP-1) secretion. Indeed, the GLP-1 response is higher after meals consumed in the early hours of the day (i.e., early timed breakfast) compared to isocaloric evening meals [18,65,66]

### 5.2. Influence of High Energy Breakfast versus High Energy Dinner on Clock Controlled Postprandial Glucose Metabolism

Based on several reports showing that the first meal of the day has a powerful resetting effect on the clock gene network [21,22,38,82,83,84,85,86,87], the temporal synchronization between breakfast with light at dawn might be critical for achieving metabolic homeostasis [19,21,22,88,89,90].

Therefore, our group explored in a crossover design whether a high-energy breakfast diet (Bdiet) versus a high-energy dinner diet (Ddiet) improves postprandial and overall glycemia in participants with T2D [18]. Bdiet consisted of three meals: high-energy and CH breakfast (700 kcal, with 54% of daily CH), medium-sized lunch (600 kcal, with 36% of daily CH), and low in energy and CH dinner (200 kcal, with only 10% of daily CH intake). The participants ate breakfast between 8:00–10:00, lunch between 13:00–15:00, and dinner between 18:00–20:00. Ddiet had a reversed plan, including a small breakfast, similar lunch, and a large energy dinner [18].

There was a significant reduction in overall glucose excursions and hunger scores in Bdiet vs. Ddiet. In parallel, postprandial insulin, C-peptide, intact GLP-1, and total GLP-1 secretion were significantly enhanced in Bdiet [18]. Figure 4.

The higher and faster insulin response, especially the first phase of insulin secretion, after breakfast, lunch, and dinner in Bdiet compared to Ddiet suggests an improvement of β-cell responsiveness and β-cell memory [91,92]. The increased overall GLP-1 in Bdiet also enhances the insulinotropic effect and may further explain the reduction of overall glycemic excursions in Bdiet [18,93]. Other reports also suggest that shifting more energy to the early hours of the day has a metabolic advantage on overall glycemia compared to diet interventions (DI), with more energy assigned to afternoon/evening hours [4,5,6,11,12,13,14,15,16,79].

### 5.3. Clock Controlled Metabolism during Postabsorptive State

The circadian clock also plays a pivotal role during the postabsorptive state. The physiological postabsorptive state is the overnight fast (i.e., 23:00–07:00). It refers to a 6–12 h period from the last meal until the next meal. During postabsorptive state the blood glucose levels are maintained mainly through hepatic glucose production [38,45,82].

After the last meal of the day, at the beginning of the nocturnal sleep, the insulin/glucagon ratio initially increases to stimulate glucose storage as glycogen in the liver [45]. Following the digestion of this meal, and as the nocturnal fasting progresses, the insulin/glucagon ratio decreases, then the liver shifts into releasing glucose into the blood (via glycogenolysis and gluconeogenesis) to maintain a constant concentration of glucose [45,81,93]. Therefore, during the overnight fast, the glucose metabolism gradually moves from an anabolic into a catabolic state [38,45,82,93].

During the postabsorptive state, the liver also takes up the free fatty acids (FFAs) released into the circulation from the lipolysis of adipose tissue to provide energy for the liver and generate ketones for use by other tissues [38,83,84]. The liver also breaks down glycogen and amino acids to generate glucose for the brain [30,31,32,38].

At the molecular level, during the overnight fast, the *BMAL1: CLOCK* complex promotes the expression of *REVERBα*, *RORα*, SIRT1, and *PGC-1α* in the liver. The glucagon-mediated cAMP-response element-binding protein (CREB) is upregulated. It stimulates the rhythms of enzymes involved in the postabsorptive nocturnal hepatic glucose production, namely the hepatic glucose 6-phosphatase (HG6-P) in the glycogenolysis pathway and the gluconeogenic enzyme, phosphoenolpyruvate carboxykinase (PEPCK) [56,57,86,94,95]. Figure 1 and Figure 3.

Therefore, the circadian clock coordinates the nocturnal oscillation of hepatic glucose output: the glycogenolysis pathway, which is more active in the first part of the overnight sleep, and the gluconeogenesis, more active during the second part of the overnight fast before waking up [56,57,86,94,95]. In addition, *REV-ERBα*, RORα, and the *BMAL1* transcription of *PPARα* and *PGC-1α* in the adipose tissue regulate the nocturnal lipolysis [52,55,96]. Figure 1.

### 5.4. Clock-Controlled Glucose Metabolism during the Transition (Switch) from Overnight Fast to Postprandial State

The circadian clock coordinates changes in insulin secretion and action to ensure appropriate substrate switching between tissues to meet metabolic needs in the transition from postabsorptive to postprandial state [38,45,97,98].

After the early-timed breakfast, the blood glucose levels across the day depend mainly on insulin-mediated muscular glucose uptake [38,98]. This switch to insulin-mediated muscular glucose uptake is crucial in order to prevent postprandial hyperglycemia after breakfast and the subsequent meals across the day [38,45,50]. Simultaneously, with breakfast consumption, the hepatic clock switches from the resting phase glycogenolysis to postprandial phase glycogenesis [4,50].

The increase of *AMPK* expression after a nocturnal fast significantly enhances GLUT-4 translocation and muscle glucose uptake once the day’s first meal is consumed, ensuring circadian changes of metabolic efficiency, thus improving morning versus evening glucose and postprandial insulin responses [7,56,86].

## 6. Effect of Fasting until Noon (Extended Postabsorptive State) on Clock Genes mRNA Expression and Regulation of Body Weight, Glucose Metabolism, Appetite, and Energy Expenditure

In the same way as eating at an inappropriate time can cause metabolic disturbances, the lack of meals during the activity phase may negatively affect the metabolism. As mentioned, skipping or delaying breakfast until noon is associated with weight gain, increased risk for T2D, and cardiovascular diseases [10,19,23,39,40,41,99].

While early-timed breakfast is a known powerful zeitgeber for clock genes in peripheral tissues, the absence of breakfast or breakfast delayed until noon has the potential to cause asynchrony of clock genes expression and disrupted regulation of metabolic outputs [10,15,20,22,44].

### 6.1. Effect of Fasting until Noon on Clock Genes mRNA Expression and Glucose, Insulin, and Incretin Responses after Subsequent Meals

Several studies documented that the absence of the day’s first meal, equivalent to a delayed breakfast until noon, leads to disrupted and blunted rhythmicity of clock gene expression in animal models [19,21,41] and humans [22]. It was also shown that fasting until noon causes an increase in the glycemic response and diminished insulin response after a subsequent meal [22,27,44,45,100,101,102]. Moreover, the delayed breakfast until noon, in the context of TRE modality consisting of fasting until noon and eating window in the afternoon and evening, has been associated with deficient control of body weight [15,16,39,40,79,80,102], increased overall glycemia [28,42,43], hunger scores and appetite hormones, i.e., ghrelin, and decreased energy expenditure [10,15,25,103,104,105].

Therefore, in healthy and T2D individuals, we explored the effect of extended postabsorptive fasting until noon versus breakfast consumption on clock gene mRNA expression and glycemic, insulin, and incretin excursions after a subsequent isocaloric lunch [22]. The participants were randomly assigned in crossover to single-test day, either to early breakfast consumption at 8:00, lunch, and dinner (YesB), or to another testing day (NoB) with only lunch and dinner but the omission of breakfast [22].

The prolonged fasting, ~16 h until noon (21:00 to 12:00), NoB acutely disrupted clock gene expression, and downregulated *AMPK*, *BMAL1*, PER1, and RORα mRNA expression before and after lunch. It was associated with significantly higher glucose, deficient and delayed insulin, and lower intact GLP-1 responses after lunch vs. YesB. In contrast, cutting the overnight fast at 8:00, with high energy breakfast on YesB day, led to a resetting effect of these key metabolic clock gene mRNA expression, significant reduction of postprandial glycemia, and enhanced and faster insulin and GLP-1 post-lunch responses [22]. Figure 5.

It has been documented that the upregulation of *AMPK*, shown in YesB day, significantly enhances GLUT-4 translocation, muscle glucose uptake, and postprandial insulin response, leading to reduced post-meal glycemic excursions [56]. However, this increased *AMPK* expression occurs only upon consuming an early-timed breakfast, ensuring metabolic efficiency and improving glucose and insulin postprandial responses [7,56,86]. Further, *AMPK* is also positively linked to *SIRT1* and its beneficial effects on insulin sensitivity, β-cell proliferation, and viability [61,86]. It suggests that the disturbed clock genes expression triggered by the absence of breakfast may be the underlying mechanism of higher glycemia, and deficient insulin, and intact GLP-1 postprandial responses [22].

In another single-day crossover study in T2D, comparing YesB versus NoB testing days, we have found that the deleterious effects of fasting until noon, i.e., increased glucose and deficient insulin and GLP-1 responses after lunch, were also observed after dinner on the NoB day [44]. Figure 6.

The omission or lack of the day’s first meal was associated with poor β-cell memory and β-cell responsiveness at the next meals [91,92]. Further, fasting until noon may induce lysosomal degradation of nascent insulin secretory granules and less β-cell secretory granule biogenesis [92], underscoring the poor and delayed postprandial insulin response to lunch and dinner on NoB day [44]. These results align with other studies showing that omission of breakfast or late TRE is associated with increased postprandial and overall glycemia [15,16,27,28,100,101,102].

### 6.2. Effect of Fasting until Noon at the Switch from Overnight Fast to Fed State on Clock-Controlled Glucose Metabolism and Skeletal Muscle Protein Synthesis and Breakdown

Hepatic glucose production (via glycogenolysis) maintains blood glucose levels during the overnight (resting) fast. After waking up at dawn and once an early-timed breakfast is consumed, the hepatic clock switches from resting phase glycogenolysis to postprandial (active phase) glycogenesis [4,50]. The omission of early-timed breakfast is a challenge that activates several surviving mechanisms, switching to alternative metabolic pathways in the liver and muscle to ensure glucose availability [20,41,47]. In response to extended fasting after waking up, once glucose levels begin to fall, the rise of glucagon levels leads to the increase of hepatic gluconeogenesis and glucose production, thus restoring blood glucose levels [20,38,41,47]. Therefore, the hepatic glucose production, which usually occurs during the overnight fast, is shifted to the early hours of the active phase [20,41,47].

At the molecular level, the omission of breakfast affects the clock genes mRNA expression and clock proteins, resulting in blunted and delayed rhythmicity of BMAL1 and REV-ERBα both in the liver and skeletal muscle [20,38,46]. In addition, the prolonged postabsorptive state implicated in the omission of breakfast activates clock gene-mediated gluconeogenesis by increasing transcriptional factors involving glucagon-mediated cAMP-responsive element-binding protein (CREB) transcription coactivator 2 (CRTC2). It reactivates the transcription of gluconeogenic genes and gluconeogenesis during the active phase to further facilitate hepatic glucose production [20,47,106,107].

The response to the extended morning fast involves massive gluconeogenesis, requiring skeletal muscle protein breakdown to provide glucogenic amino acids (mainly alanine) as a substrate to replace liver glycogen and to supply glucose to the brain [20,30,31,32,41,46,47,107]. Although the transportation of alanine to the liver and its conversion to glucose via gluconeogenesis provides an effective mechanism to maintain blood glucose levels during an extended-postabsorptive state, it uses a large protein reserve from skeletal muscle on the day when breakfast is omitted [38,106,108]. It was shown in mice that the omission of breakfast, besides delaying clock-gene expression in the liver, adipose tissue, and muscle, is associated with a risk of obesity and sarcopenia with significant loss of muscular mass [41].

Long-term omission or delayed breakfast until noon may increase the risk of losing skeletal muscle mass and sarcopenia in those who frequently skip breakfast [30,31,32]. Therefore, early breakfast consumption aligned with the circadian transition from a nocturnal fasting state to a fed state might be critical for the appropriate regulation of glucose metabolism while preserving muscle mass

### 6.3. Influence of Fasting until Noon on Circadian Clock-Controlled Regulation of Body Weight, Appetite, and Energy Expenditure

Dietary interventions (DI) for weight loss typically focus on the magnitude of the energy deficit. However, growing evidence shows that the timing of fasting and food intake can influence the metabolic response [10,13,14,15,16]. The omission of or delayed breakfast until noon is associated with weight gain, which frequently cannot be explained by differences in reported caloric intake [19,23,39,40].

Previous studies have shown that an early-timed high-energy breakfast was associated with more significant weight loss than a high-energy dinner [10,15,17,102,104,105]. Likewise, the modality of early TRE, consisting of early-timed breakfast and fasting in the late hours of the day, is more efficient for achieving weight loss and glycemic control than fasting until noon and eating later in the day [15,79].

Fasting in the morning is apparently less beneficial for weight loss than fasting in the evening [10,15,16,79,102]. Indeed, breakfast omission has been proposed as a key modifiable risk factor for obesity [102,103,104,105]. The underlying mechanism linking breakfast omission with obesity and less-efficient weight loss is unknown. However, higher energy expenditure and diet-induced thermogenesis (DIT) in the morning versus evening may play a role in the beneficial effect of early-timed breakfast on weight loss outcomes [10,34,35,36,37].

The endogenous circadian clock regulates energy expenditure (EE), and in humans, EE is at its lowest levels during the biological night. The respiratory quotient (RQ), reflecting macronutrient utilization, also varies by circadian phase and is highest in the early morning [80,81]. Several studies have documented that diet-induced thermogenesis (DIT) is higher in the morning than in the evening [34,35,36,37]. Hence, isocaloric meals lead to a 2.5-fold higher DIT in the morning than in the evening. Therefore, consuming a large dinner meal rather than an extensive breakfast may promote obesity [36]. However, it remains unclear whether a diet intervention (DI) with a delayed breakfast until noon influences diet-induced thermogenesis and 24-h energy expenditure [104,105].

Therefore, a recent randomized crossover study was conducted to determine whether fasting until noon and late evening meals is associated with decreased energy expenditure and changes in appetite, and whether molecular pathways in adipose tissues are involved [25]. The participants were assigned in a crossover design to two isocaloric diet interventions (DI), with three meals but different meal timing. In the “early meal” DI period, participants ate early breakfast, lunch, and dinner at about 09:00, 13:00, and 17:00. In the “late-meal” DI, the participants skipped breakfast, began eating lunch at 13:00, ate dinner at 17:00 and added a late supper at 21:00, to compensate for the skipped breakfast [25]. The study showed that fasting until noon and late-meal DI was associated with significantly increased hunger VAS scores, decreased leptin, and increased ghrelin levels [25]. These findings align with previous reports showing that consuming a morning-loaded diet significantly lowers hunger scores [13,17,18,29,33].

Adipose tissue gene expression analyses during the fasting until noon DI period showed alterations in lipid metabolism pathways, e.g., TGF-β signaling and modulation of receptor tyrosine kinases, in a direction consistent with increased adipogenesis and decreased lipolysis [25].

Further, fasting until 13:00 in the late-meal DI protocol led to a significant decrease in energy expenditure compared to the early-meal DI with early-timed breakfast [25]. Although skipping breakfast in the late-meal DI protocol was associated with decreased energy expenditure, we cannot rule out that this effect was also related to the late-timed supper at 21:00. However, this seems unlikely since the early-timed breakfast triggers a greater increase in DIT compared to the evening meal [34,35,36,37].

The decreased energy expenditure in the fasting until noon DI schedule suggests that skipping breakfast may increase the risk of obesity and reduce the efficacy of DIs aimed at weight loss [11,12,19,23,25,39,40,102,103,104,105]. The early-timed high-energy breakfast may assist in weight loss DI through appetite suppression [13,17,18,29,33]. Further, it provides a mechanistic explanation of greater efficiency of the weight loss outcomes from the DI with more energy shifted to the early hours of the day [10,11,12,17,18,29,33,109].

## 7. Conclusions

The circadian clock controls the diurnal variation of most metabolic processes through dynamic bidirectional synchronization between fasting/feeding cycles and light input. It regulates the glucose and other metabolic processes during fasted and fed states and during the switch from an overnight fast to a postprandial fed state.

The early-timed breakfast at dawn shortly after waking up has a powerful resetting effect on the clock gene network and is crucial for transitioning from an overnight fast to a fed state. Glucose, insulin, GLP-1, and other postprandial signals rising after early-timed breakfast upregulate clock genes and transcription factors, improving glucose and other metabolic processes throughout the remainder of the active phase. In contrast, postponing the first meal of the day creates a metabolic challenge at the hour when the clock gene network is anticipating early-timed meal consumption.

In this review, we have shown that the misalignment of the feeding fasting cycle, and particularly the lack of early-timed breakfast, leads to deleterious effects on clock gene expression, disrupted regulation of body weight, and increased overall glycemia and postprandial glucose, along with deficient insulin and incretin responses after subsequent meals. Skipping breakfast activates skeletal muscle protein breakdown as part of a survival mechanism to provide glucogenic amino acids to the liver. Moreover, the delayed breakfast until noon increases hunger scores and even may cause a lowering effect on energy expenditure. It may further increase the risk of obesity and reduce the efficacy of DI aimed at weight loss.

Although many more studies are needed to explore the underlying mechanism of these deleterious effects of breakfast omission, it seems that shifting caloric intake toward the beginning of the active phase (i.e., early-timed breakfast) may confer a metabolic advantage, especially in the context of weight loss and overall glycemia, by coordinating food intake to coincide with optimal circadian timing.

## Figures and Tables

**Figure 1 ijms-24-07154-f001:**
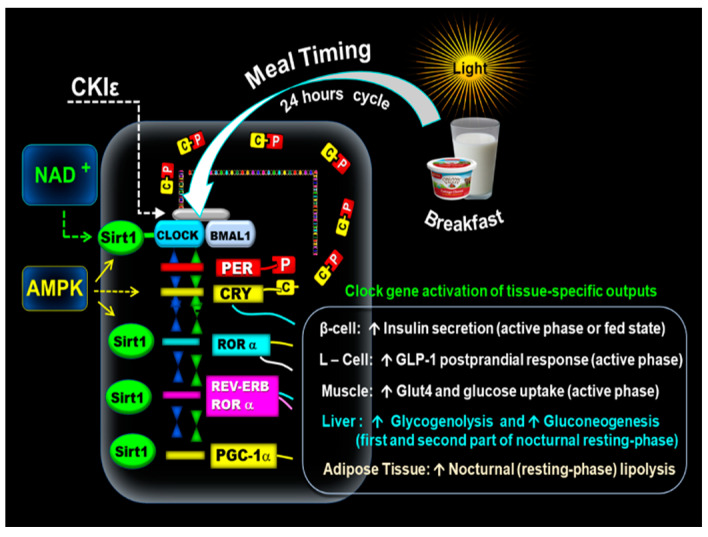
Mechanism of the molecular clock. The above illustration shows that the early-timed breakfast at dawn activates the CLOCK: BMAL1 complex associated with SIRT1 and the transcription of PERs and CRYs. The resulting proteins PER (P) and CRY (C) form PER: CRY (C-P) dimers in the cytoplasm. Subsequently, PER: CRY dimers translocate back to the nucleus to repress CLOCK: BMAL1. The blockage of CLOCK: BMAL1 is reversed by casein kinase I epsilon (CKIε). NAD+ activates SIRT1, which interacts with the CLOCK: BMAL1. In addition, AMPK positively interacts with SIRT1. CLOCK: BMAL1-driven transcription of PERs, CRYs, REV-ERBα, RORα genes, and PGC-1α promotes the expression of tissue-specific clock-controlled genes. It leads to the upregulation of β-cell insulin secretion, L-cell postprandial incretin GLP-1 response, and the increase of GLUT4 activity and muscle glucose uptake. The clock gene-driven nocturnal hepatic glucose production promotes glycogenolysis and gluconeogenesis in the first and second part of the nocturnal resting phase. PGC-1α plays a role in lipid metabolism (lipogeneses and nocturnal lipolysis.

**Figure 2 ijms-24-07154-f002:**
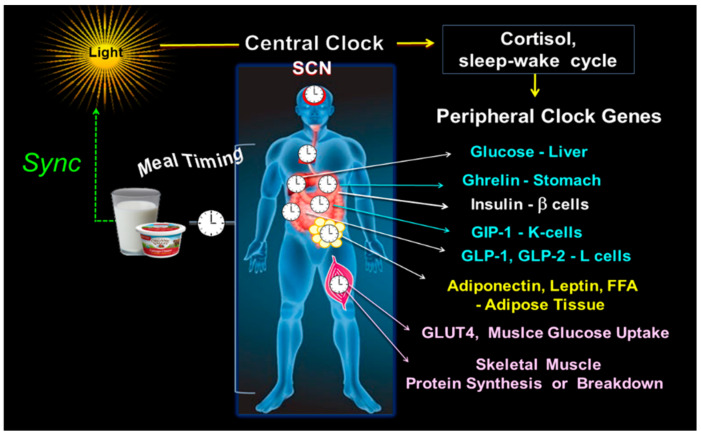
Central and peripheral clocks alignment. The illustration shows the central clock in SCN, responsive to light input, and peripheral clocks, activated by meal timing, both in synchrony. On the right side are some examples of the peripheral CGs outputs, i.e., the hepatic glucose production; ghrelin secretion in the stomach; b-cell insulin secretion; the intestinal K-cell secretion of GIP and L-cell secretion of GLP-1 and GLP-2; leptin and adiponectin in adipose tissue; GLUT4 driven muscle glucose uptake; and the muscle protein breakdown/synthesis.

**Figure 3 ijms-24-07154-f003:**
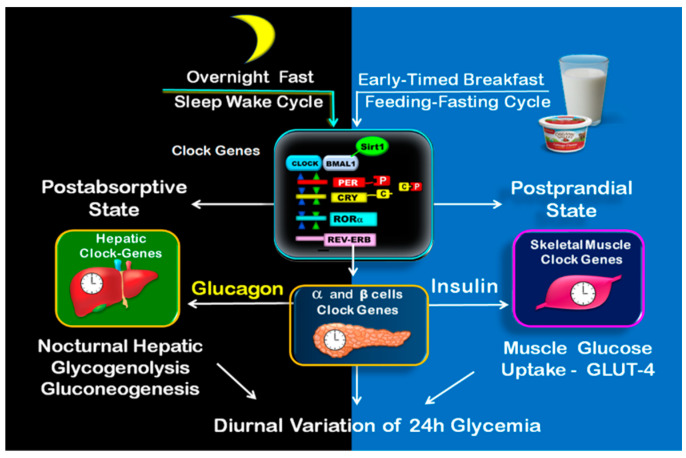
Clock-controlled postprandial and postabsorptive glucose metabolism. The illustration shows on the left side the clock-controlled glucose metabolism during overnight fast (postabsorptive state), consisting of nocturnal hepatic glycogenolysis and gluconeogenesis. The postprandial state is shown on the right side. It begins upon early-timed breakfast and consists mainly of muscle glucose uptake.

**Figure 4 ijms-24-07154-f004:**
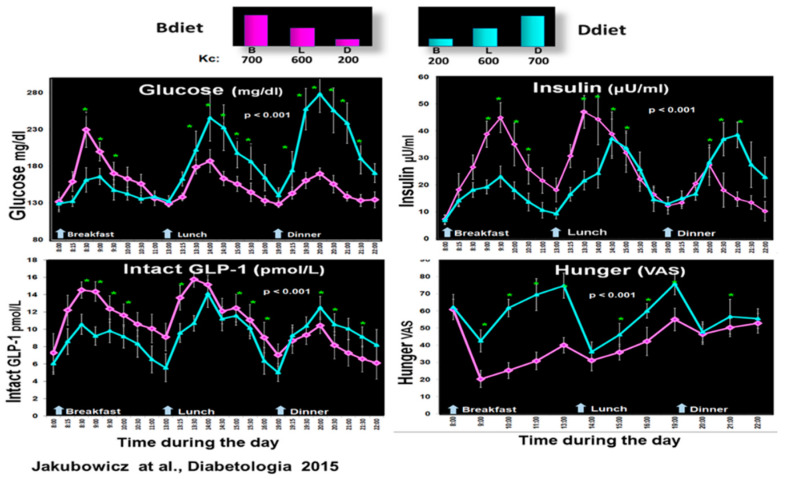
Effect of early-timed high energy breakfast diet (Bdiet) versus high energy dinner diet (Ddiet) on postprandial glycemia, insulin, intact GLP-1, and hunger scores in T2D. In the upper part is shown the caloric content for Breakfast (B), Lunch (L), and Dinner (D) on Bdiet day (pink) and Ddiet day (blue). The line charts show an all-day graph for overall postprandial glycemia, insulin, intact GLP-1, and hunger VAS in Bdiet vs. Ddiet group. * *p* < 0.05 “Reproduced and adapted with permission” [18].

**Figure 5 ijms-24-07154-f005:**
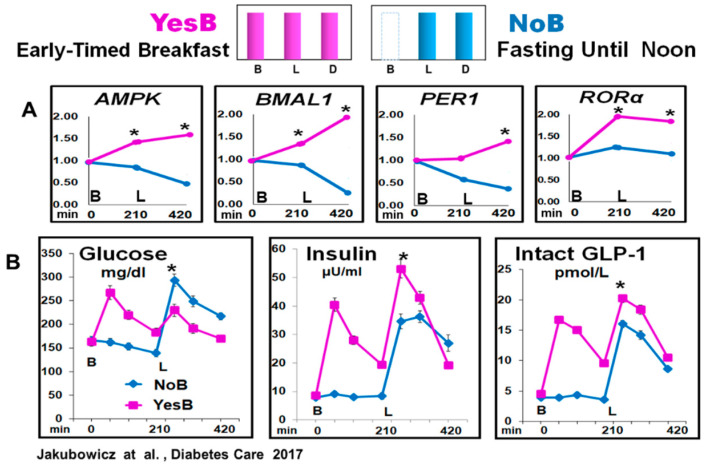
Clock Genes (AMPK, BMAL1, PER1 and RORα) mRNA expression and glucose, insulin, and intact GLP-1 blood levels, at fasting, before lunch and after lunch in YesB and NoB day in T2D. (**A**) Clock gene expression: Blood samples were collected in YesB (purple) and NoB day (blue), at fasting (time point 0 min), before lunch (time point 210 min), and 3.5 h after lunch (time point 420 min). Asterisks denote statistical differences (*p* < 0.05) between time points 210- and 420 min. B: Line charts of glucose, insulin, and intact GLP-1 postprandial responses in YesB and NoB days. Breakfast (**B**) was given to the YesB group at time point 0. Lunch (L) was given to both groups at a time point of 210 min. Asterisks denote statistical differences between YesB and NoB at a specific time point. Data are means ± SE. “Reproduced and adapted with permission” [22].

**Figure 6 ijms-24-07154-f006:**
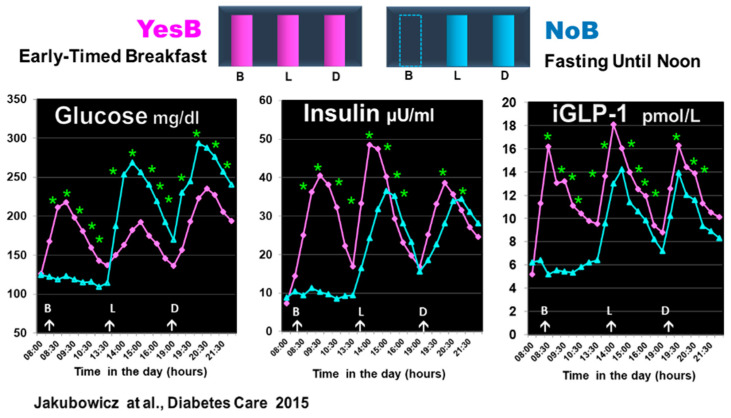
Effect of prolonged fast until noon (NoB) vs. breakfast consumption (YesB) on glucose, insulin, and intact GLP-1 (iGLP-1). The blood samples were taken at fasting and postprandial responses after breakfast or no breakfast, lunch, and dinner at the same time points. Statistics were between the NoB and the YesB. Data are means ± SE. Asterisks denote statistical differences between YesB and NoB at a specific time point. * *p* < 0.001. B: Breakfast; L: Lunch; D: Dinner. NoB: purple lines; YesB: blue lines “Reproduced and adapted with permission” [44].

## Data Availability

Not applicable.

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
