# Peer review of "Influence of Fasting until Noon (Extended Postabsorptive State) on Clock Gene mRNA Expression and Regulation of Body Weight and Glucose Metabolism"

_ijms, 2023, doi:10.3390/ijms24087154_

Round 1

Reviewer 1 Report

The review article is well written and explained all the aspects of delayed breakfast.  Deleterious effect of fasting until noon includes altered clock gene expression and body weight gain, hyperglycemia, insulin resistance etc.

There are few things that can be included.  

1.       Figure 1: Follow a uniform pattern – either by mentioning tissue/ cell types with the pathways activated or inhibited OR by using gene names that are activated or inhibited. And provide which pathways/genes are upregulated and which pathways/genes are inhibited either by using up/down arrows or using sharp arrows /blunt arrows.

2.       Line 106 – 113 & Figure 1: Kindly provide clear statements on which are the pathways activated in nocturnal and which are activated in postprandial conditions.   

3.       Figure 3 (Left side): Overnight fast reduces insulin production. The figure has “Insulin” in the overnight fast side (Left side).  Kindly make it clear (use arrows for increase / decrease).

4.       Section 4.3: Potential benefits of early-timed breakfast: Kindly include a brief note about the studies on lipid profiles as well.

Author Response

Dear  Reviewer 1

We very much appreciate your favorable opinion of the manuscript and are grateful for your proposed changes. All corrections  are  addressed  in the new draft of the manuscripts as follows:

  1. Figure 1: Follow a uniform pattern – either by mentioning tissue/ cell types with the pathways activated or inhibited OR by using gene names that are activated or inhibited. And provide which pathways/genes are upregulated and which pathways/genes are inhibited either by using up/down arrows or using sharp arrows /blunt arrows.

Response: To address your suggestions to follow a uniform pattern of the clock-controlled actions, we wrote, schematically, a list of the specific tissue/cells targeted by the clock genes transcription and the resulting outputs in each one of these tissue/cells. We hope the tissue-specific actions driven by the circadian clocks are now easier to understand. The previous Figure 1 and the corrected Figure 1 are shown in the corrected manuscript from Line 94 to 95

  1. Line 106 – 113 (B) & Figure 1 (A): Kindly provide clear statements on which are the pathways activated in nocturnal and which are activated in postprandial conditions. 

Response:  A. The legend of Figure 1 has been modified at the end (in red) according to the changes in Figure 1, describing the clock-driven tissue/cells specific targets and outputs. ( lines 100-103).

B: Regarding the reviewer's request to provide clear statements on which pathways are activated in nocturnal and which are activated in postprandial conditions. This paragraph was rewritten (in red, now in lines 110 to 117), emphasizing better which pathways are activated in the active phase (fed state) and the pathways activated during the nocturnal resting phase.

  1. Figure 3 (Left side): Overnight fast reduces insulin production. The figure has “Insulin” in the overnight fast side (Left side).  Kindly make it clear (use arrows for increase / decrease).                                Response: We thank the reviewer for this observation of Figure 3. Given that the insulin is reduced during the nocturnal fast, and blood glucose is maintained through liver glucose production driven by glucagon, we eliminated the word "insulin" from the overnight fast (left side) to avoid confusion. This difference in glucose metabolism is also widely explained in the section 5.3, "Clock controlled metabolism during postabsorptive state" (lines 300 to 328).  The previous  Figure 3 and the  corrected Figure 3 are displayed in the corrected manuscript  (line 238)
  2. Section 4.3: Potential benefits of early-timed breakfast: Kindly include a brief note about the studies on lipid profiles as well.              Response: To include more data on the effect of early timed breakfast on lipid profile, we added paragraph discussing studies  (Ref 79 and 80) on the influence of eating versus skipping breakfast on plasma lipids in the context of TRE  (in red lines 196 to 205).  We  also  added  in the same section  a description of the effect of Bdiet versus Ddiet  on plasma triglyceride levels  (in red, lines 214 and  215).

Reviewer 2 Report

This is an extremely important review, revising the literature on the time food restriction, clock functioning, and obesity. The authors showed that the omission of breakfast may lead to disruption of the clock gene expression, dysregulation of glycemia, deficient insulin and incretin responses after subsequent meals, and eventually obesity.

Only some minor comments

Line 231 CLOCK in italic

Line 359 glucose with lowercase

Line 386- 391- remove italics

Author Response

Dear Reviewer 2

Thank you very much for such positive comments on the manuscript. 
The errors you addressed have been corrected as follows:

Line 231 CLOCK in italic → corrected  in red, now in line 253

Line 359 glucose with lowercase → corrected  in  red, now in line 382

Line 386- 391- remove italics →corrected  in  red, now in lines 407-409
